# Dynamics of pulsatile activities of arcuate kisspeptin neurons in aging female mice

Teppei Goto*, Mitsue Hagihara, Kazunari Miyamichi*

Laboratory for Comparative Connectomics, RIKEN Center for Biosystems Dynamics Research, Kobe, Japan

**Abstract** Reproductive senescence is broadly observed across mammalian females, including humans, eventually leading to a loss of fertility. The pulsatile secretion of gonadotropin-releasing hormone (GnRH), which is essential for gonad function, is primarily controlled by kisspeptin neurons in the hypothalamic arcuate nucleus (ARC$^{kiss}$), the pulse generator of GnRH. The pulsatility of GnRH release, as assessed by the amount of circulating gonadotropin, is markedly reduced in aged animals, suggesting that the malfunctions of ARC$^{kiss}$ may be responsible for reproductive aging and menopause-related disorders. However, the activity dynamics of ARC$^{kiss}$ during the natural transition to reproductive senescence remain unclear. Herein, we introduce chronic in vivo Ca$^{2+}$ imaging of ARC$^{kiss}$ in female mice by fiber photometry to monitor the synchronous episodes of ARC$^{kiss}$ (SEs$^{kiss}$), a known hallmark of GnRH pulse generator activity, from the fully reproductive to acyclic phase over 1 year. During the reproductive phase, we find that not only the frequency, but also the intensities and waveforms of individual SEs$^{kiss}$, vary depending on the stage of the estrus cycle. During the transition to reproductive senescence, the integrity of SEs$^{kiss}$ patterns, including the frequency and waveforms, remains mostly unchanged, whereas the intensities tend to decline. These data illuminate the temporal dynamics of ARC$^{kiss}$ activities in aging female mice. More generally, our findings demonstrate the utility of fiber-photometry-based chronic imaging of neuroendocrine regulators in the brain to characterize aging-associated malfunction.

*For correspondence:
teppei.goto@riken.jp (TG);
kazunari.miyamichi@riken.jp
(KM)

**Competing interest:** The authors declare that no competing interests exist.

## Editor's evaluation

The reviewers agreed that your studies demonstrating an age-dependent dynamic change in the activity of key neurons (Kisspeptin neurons) regulating reproductive functions will be of wide interest. The work provides functional links regarding the transition to reproductive senescence characteristic of aging. The use of chronic imaging of identified neurons is also an important advance.

## Introduction

Reproductive aging occurs broadly across mammalian females, eventually leading to loss of fertility (**Kermath and Gore, 2012**). In humans, women undergo the perimenopausal transition to reproductive senescence, in which the menstrual cycles become irregular in length and timing before eventually ceasing at approximately 51 years of age (**Gore, 2015**). As this transition is associated with a risk of metabolic, physical, and mental disorders (**Greendale et al., 1999**), it is essential to understand the molecular and cellular mechanisms of reproductive aging for not only reproductive medicine, but also women's health in general.

Reproductive function in females is primarily controlled by coordinated interactions among the three layers of the hypothalamus–pituitary–ovary axis (**Gore, 2015**; **Kermath and Gore, 2012**). At the core, GnRH-producing neurons in the anterior hypothalamus provides the driving force of the reproductive system. GnRH release occurs in a pulsatile manner at intervals of 30 min to several

hours, depending on the species and reproductive (estrus) cycle (*Herbison, 2018*), and this pulsatility is essential for reproductive functions (*Belchetz et al., 1978*). GnRH released at the median eminence acts on the anterior pituitary gonadotrophs, which then release luteinizing hormone (LH) and follicle-stimulating hormone (FSH) into the peripheral circulation. Pulsatile (tonic) LH and FSH release facilitate follicular development and steroidogenesis in the ovary. The sex steroid hormones then exert feedback control of the hypothalamic GnRH system and pituitary gonadotrophs. Although age-related changes can occur in each of these layers, classical evidence in rodent models suggests that modulation of GnRH neurons may be a major driving force of reproductive senescence. For example, pulsatile LH release as a proxy of GnRH activities decreases with age in rats (*Scarbrough and Wise, 1990*; *Wise et al., 1988*). As such, the preovulatory LH surge is delayed or attenuated with aging (*Matt et al., 1998*) with decreased c-Fos expression, a marker of neural activation, in the GnRH neurons (*Rubin et al., 1994*). Together with a classical transplantation study showing the presence of viable follicles in aged rodents (*Krohn, 1962*), these studies suggest that reproductive senescence is mediated by the neuroendocrine system independent of the depletion of the follicular reserve.

How is the homeostatic regulation of the GnRH system altered by aging? Of many candidates for GnRH regulation, kisspeptin, a neuropeptide crucial for puberty (*d'Anglemont de Tassigny et al., 2007*; *Lapatto et al., 2007*; *Uenoyama et al., 2015*), is thought to play a dominant role (*Kermath and Gore, 2012*). There are two populations of kisspeptin neurons in the rodent hypothalamus: one in the arcuate nucleus (ARC$^{kiss}$), which acts as the GnRH pulse generator (*Clarkson et al., 2017*; *Nagae et al., 2021*), and the other in the anteroventral periventricular nucleus (AVPV$^{kiss}$), which functions as the GnRH surge generator (*Clarkson et al., 2008*; *Matsuda et al., 2019*; *Piet et al., 2018*; *Uenoyama et al., 2015*). Previous studies have shown a decrease in kisspeptin signaling in both ARC$^{kiss}$ (*Kunimura et al., 2017*) and AVPV$^{kiss}$ (*Lederman et al., 2010*; *Neal-Perry et al., 2009*) during aging in rats. Thus, malfunctions of the kisspeptin system may underpin the transition to reproductive senescence.

One critical step to characterizing the kisspeptin system for reproductive aging and menopause-related disorders (*Padilla et al., 2018*) is to visualize directly the dynamics of neural activities of kisspeptin neurons during aging. Until recently, activities of the GnRH pulse generator could only be indirectly assessed by repetitive measurement of circulating LH level (*Steyn et al., 2013*), which is not only labor-intensive but also stressful to animals not fully suited for chronic analysis. Recent seminal studies have established cell-type-specific chronic Ca$^{2+}$ imaging of ARC$^{kiss}$ activities under a free-moving awake condition in mice by using fiber photometry (*Clarkson et al., 2017*; *Han et al., 2019*; *McQuillan et al., 2019*). ARC$^{kiss}$ shows remarkable pulsatile activities, termed synchronous episodes of elevated Ca$^{2+}$ (hereafter referred to as SEs$^{kiss}$ for simplicity). SEs$^{kiss}$ are well correlated with pulsatile LH secretion (*Clarkson et al., 2017*), tightly regulated by gonadal negative feedback signals mediated by sex steroid hormones (*McQuillan et al., 2022*), and undergo dynamic changes in frequency depending on the stage of the estrus cycle (*McQuillan et al., 2019*). However, the utility of this system is currently limited to only reproductive-phase or gonadectomized mice. It remains unknown how this system can facilitate the study of reproductive senescence by long-term imaging of SEs$^{kiss}$ dynamics. This motivated us to apply fiber photometry to monitor SEs$^{kiss}$ from the fully reproductive to acyclic phase over 1 year.

## Results

### Dynamics of SEs$^{kiss}$ in the regular estrus cycle at the reproductive phase

Before characterizing SEs$^{kiss}$ during aging, we revisited the detailed patterns and waveforms of SEs$^{kiss}$ in the reproductive phase. We first established 7 day chronic imaging of ARC$^{kiss}$ of cyclic *Kiss1-Cre* (*Gottsch et al., 2011*) female mice at age 4–6 months. An optical fiber was implanted above the ARC of *Kiss1-Cre* female mice that had been injected with an adeno-associated virus (AAV) driving Cre-dependent GCaMP6s (*Chen et al., 2013*) into the ARC. Their estrus cycle stages were determined by vaginal cytology (*Byers et al., 2012*; *Caligioni, 2009*), and Ca$^{2+}$ imaging was performed twice per day for 6 h each in the light and dark period with 6 h inter-imaging intervals. Estrus cycle and Ca$^{2+}$ imaging from ARC$^{kiss}$ were monitored for seven consecutive days. We detected sharp photometric peaks highly resembling the previously reported SEs$^{kiss}$ (*Clarkson et al., 2017*; *Figure 1A* and Materials and methods). The number of SEs$^{kiss}$ was dynamically altered in various stages of the estrus cycle and light and dark periods (*Figure 1B*). To align data from multiple animals, we utilized the fact that SEs$^{kiss}$

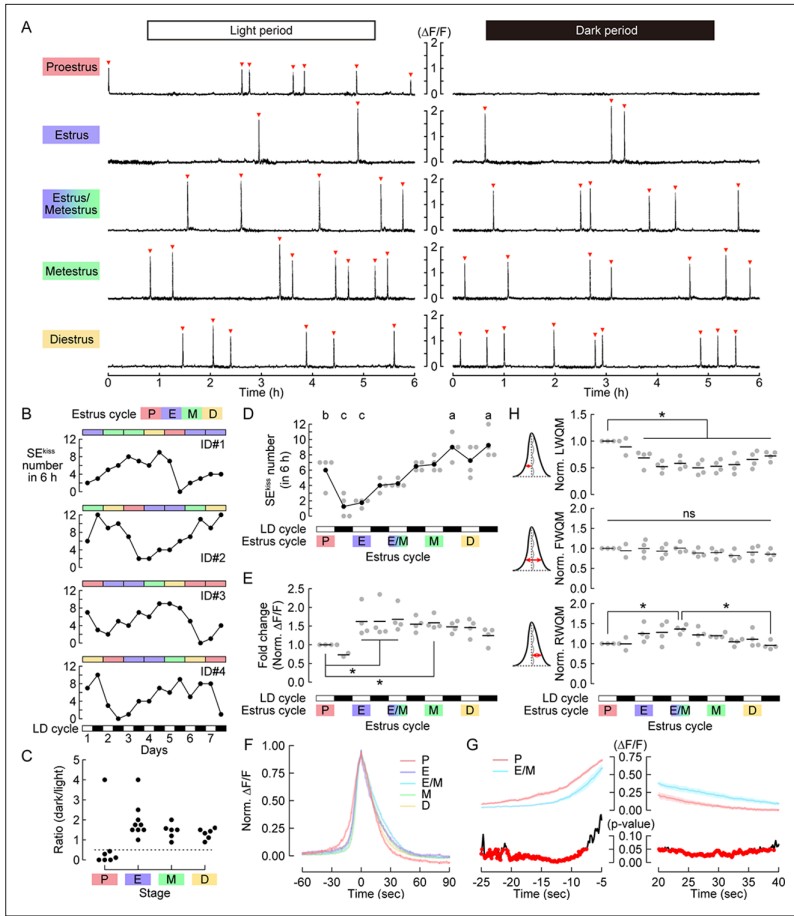

**Figure 1.** Dynamics of synchronous episodes of ARC$^{kiss}$ (SEs$^{kiss}$) in the regular estrus cycle of the reproductive phase. (**A**) Representative 6 h photometry raw data of SEs$^{kiss}$ in each stage of the estrus cycle. Red arrowheads indicate SEs$^{kiss}$. (**B**) Numbers of SEs$^{kiss}$ (lower: line plots) along with the estrus cycle stages (upper: color bars) for 7 days (n=4 animals). SEs$^{kiss}$ were recorded for 6 h in both light and dark (LD) periods. (**C**) The ratio of SEs$^{kiss}$ in each dark period normalized to those in the preceding light period. Dotted line = 0.5. (**D**) Numbers of SEs$^{kiss}$ per 6 h in 5 days along with the estrus cycle anchoring at proestrus as Day 0. Different letters (a–c) in the upper part of the graph denote significant differences at p<0.05 by one-way repeated measures ANOVA followed by the Tukey–Kramer post hoc test. (**E**) Intensities of SEs$^{kiss}$ as measured by the ΔF/F values in each estrus stage normalized to that in the light period of the proestrus. *p<0.05 by one-way repeated measures ANOVA followed by the Tukey–Kramer post hoc test. (**F**) Averaged traces of SE$^{kiss}$ waveforms in the light period of each estrus stage. (**G**) Top, averaged traces magnifying the onset and offset of SEs$^{kiss}$ in the proestrus and estrus/metestrus stages in the light period. Data are expressed as mean ± SEM. Bottom, p-value by *t*-test. Red dots represent p-values <0.05. (**H**) Quantifications of detailed parameters of SE$^{kiss}$ waveforms in each estrus stage: the normalized LWQM, FWQM, and RWQM. *p<0.05 by one-way repeated measures ANOVA followed by the Tukey–Kramer post hoc test. P: proestrus, E: estrus, M: metestrus, D: diestrus.

The online version of this article includes the following figure supplement(s) for figure 1:

**Figure supplement 1.** Inter-pulse intervals and waveforms of synchronous episodes of ARC$^{kiss}$ (SEs$^{kiss}$) in each estrus stage.

are strongly suppressed by a high level of progesterone just before ovulation in the proestrus stage (***McQuillan et al., 2019***). Indeed, when the number of SEs$^{kiss}$ in the dark period was normalized to that of the preceding light period, proestrus just after the diestrus stage showed a stereotyped drop (<0.5) compared with other stages of the estrus cycle (***Figure 1C***). Based on this profile, the estrus cycle was aligned starting from the proestrus stage. The number of SEs$^{kiss}$ during the estrus cycle was lowest at the dark period of the proestrus, and then gradually increased as the estrus cycle advanced, reaching the highest values at the dark period of the metestrus and diestrus (***Figure 1D***). As expected from this frequency of SEs$^{kiss}$, the inter-peak intervals showed the opposite trend (***Figure 1—figure supplement***

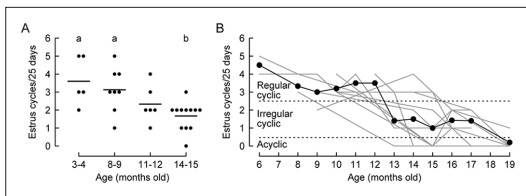

**Figure 2.** Decrease in estrus cycle frequency during the transition to reproductive senescence. (**A**) Number of estrus cycles per 25 days at 3–4 (n=5), 8–9 (n=8), 11–12 (n=6), and 14–15 (n=12) months in old wild-type C57BL/6 mice. Different letters (a, b) in the upper part of the graph denote significant differences at p<0.05 by one-way ANOVA followed by the Tukey–Kramer post hoc test. (**B**) Chronic monitoring of estrus cycle frequency from the individuals used for Ca²⁺ imaging of synchronous episodes of ARC^kiss (SEs^kiss) (C57BL/6 *Kiss-Cre* mice with AAV-mediated GCaMP6s expression). Black: average number of estrus cycles, gray: individual data (n=12 animals in total).

1A). We also found that the peak height of SEs^kiss normalized to that in the light period of proestrus fluctuated during the regular estrus cycle, with the highest (approximately 1.5-fold) at the light period 2 days after proestrus (*Figure 1E*). These data show that not only the frequency (*McQuillan et al., 2019*) but also the intensities of individual SEs^kiss vary precisely depending on the stage of the estrus cycle. The higher peak during metestrus may be correlated with a higher amount of kiss-peptin released to GnRH neurons to facilitate more LH/FSH secretion (*Czieselsky et al., 2016*).

We then quantified detailed waveforms of individual SEs^kiss in each stage of the estrus cycle (*Figure 1F*, *Figure 1—figure supplement 1B*). Although the overall shape was similar when the peak was aligned, the onset and offset in proestrus significantly advanced compared with the estrus/metestrus stage 2 days after proestrus (*Figure 1G*). To confirm this trend further, we quantified the left half-width at quarter maximum (LWQM) and right half-width at quarter maximum (RWQM). Whereas the full-width at quarter maximum (FWQM) was constant across all stages of the estrus cycle, LWQM in the proestrus stage was significantly longer than that in the other stages (*Figure 1H*). RWQM in the proestrus stage tended to be shorter, while that in estrus/metestrus was significantly longer. Variations in waveforms correlated with the peak height: when the peak was higher, the upslope was steeper and the downslope was gentler. This difference likely reflects the activity dynamics of individual ARC^kiss during each SE^kiss (see Discussion).

## Reduced estrus cyclicity with aging in *Kiss1-Cre* mice

Next, we characterized the progress of reproductive decline in our experimental animals (*Kiss1-Cre* females in C57BL/6 background). In rodents, the most revealing marker of the reproductive stage is the estrous cyclicity determined by a vaginal smear, which is an indicator of the ovarian and neuroendocrine environments (*Byers et al., 2012*; *Caligioni, 2009*). A longitudinal study of estrus cyclicity in aging rodents revealed an age-related reduction of cycle frequency and changes in cytology (*LeFevre and McClintock, 1988*; *Nelson et al., 1982*). Middle-aged female rodents can be classified into three groups: regularly cycling, irregularly cycling, and acyclic categories. We first confirmed that the number of estrus cycles per 25 days of wild-type C57BL/6J females was gradually decreased in an age-dependent manner (*Figure 2A*), generally consistent with a preceding study (*Nelson et al., 1982*). A similar trend was observed in *Kiss1-Cre* female mice that had been injected with an AAV driving Cre-dependent GCaMP6s into the ARC with an optic fiber implanted above it, indicating that the genetic manipulation and fiber photometry setting did not alter the reproductive decline process (*Figure 2B*, n=12). The average number of estrus cycles per 25 days substantially declined between ages 12–13 months, supporting the notion that female mice begin to show irregular cyclicity at around 13 months of age. Based on these data, we set the threshold at three estrus cycles per 25 days and classified middle-aged female mice into one of three groups: regular cyclic (three or more), irregular cyclic (two or one), or acyclic (zero).

## Dynamics of SEs^kiss during the transition to reproductive senescence

To characterize SEs^kiss during aging, we first chronically monitored the same female mice (n=2) from age 6 months (fully reproductive) to age 15–18 months (after reproductive senescence). To this end, we repeatedly conducted 7 day chronic imaging of ARC^kiss together with a vaginal smear-based analysis of the estrus cycle (*Figure 3* and *Figure 3—figure supplement 1*). At a glance, the frequency of SEs^kiss was not drastically changed, whereas the intensity of SEs^kiss exhibited a decline across the transition to reproductive senescence (*Figure 3A and B*). Quantitative analysis showed that the frequency

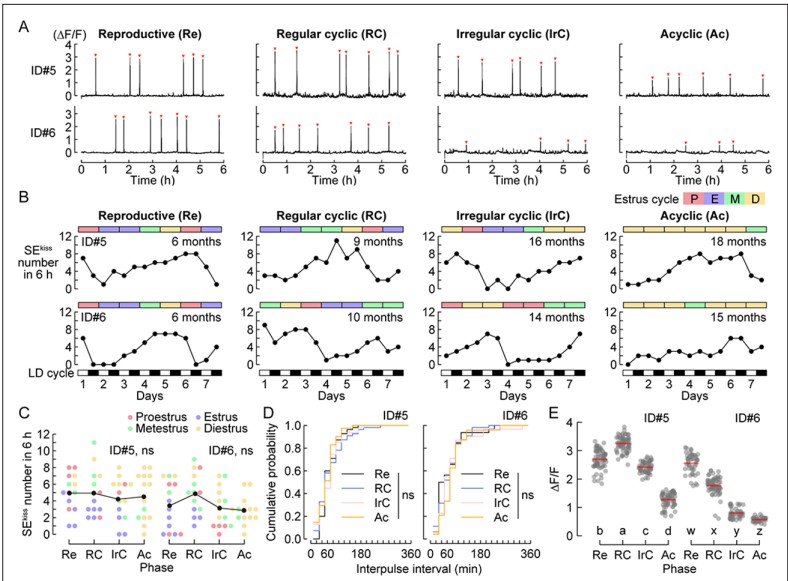

**Figure 3.** Chronic monitoring of synchronous episodes of ARC$^{kiss}$ (SEs$^{kiss}$) from the reproductive to acyclic phase. (**A**) Representative photometry raw data of SEs$^{kiss}$ in the diestrus stage from the reproductive to acyclic phase (n=2 animals). Red arrowheads indicate SEs$^{kiss}$. (**B**) Numbers of SEs$^{kiss}$ in the estrus stages (color bars) for 7 days from the reproductive to acyclic phase. SEs$^{kiss}$ were recorded for 6 h in both light and dark periods. (**C**) Number of SEs$^{kiss}$ in individual estrus stages (shown in different colors) of individual female mice. ns: not significant by one-way ANOVA. (**D**) Cumulative probability of inter-pulse interval. ns: not significant by the Kolmogorov–Smirnov test. (**E**) Peak height as assessed by ΔF/F of SEs$^{kiss}$ from the reproductive to acyclic phase. Different letters (a–d, w–z) in the lower part of the graph denote significant differences at p<0.05 by the Kruskal–Wallis test followed by the Mann–Whitney $U$ test. Re; reproductive, RC; regular cyclic, IrC; irregular cyclic, Ac; acyclic.

The online version of this article includes the following figure supplement(s) for figure 3:

**Figure supplement 1.** Raw data of the estrus cycle, related to *Figure 3*.

**Figure supplement 2.** Waveforms and intensities of synchronous episodes of ARC$^{kiss}$ (SEs$^{kiss}$), and gene expression analysis.

of SEs$^{kiss}$ as assessed by the average number per 6 h was constant (*Figure 3C*), with no difference in the cumulative probability of inter-peak intervals among data set from age 6 months (fully reproductive) to age 15–18 months (acyclic in aging) (*Figure 3D*). Notably, the elevated frequency of SEs$^{kiss}$ generally observed in ovariectomized females (*Clarkson et al., 2017*), which is thought to indicate a drastic decline of sex steroid hormone-mediated negative feedback (*Padilla et al., 2018*), was not found in the natural transition to reproductive senescence. We also found that the waveforms of individual SEs$^{kiss}$ were grossly unchanged from the reproductive to acyclic phase, with only slight individual differences found in the aging phases (*Figure 3—figure supplement 2A and B*). In contrast to the relatively stable frequency of SEs$^{kiss}$, quantitative analysis showed that the peak height as assessed by ΔF/F values was significantly decreased during the transition to the acyclic phase in these two females (*Figure 3E*).

For further quantification, we collected more photometry data from 13 to 16 month-old *Kiss1-Cre* female mice and classified them into three groups: regular cyclic (n=4), irregular cyclic (n=3), and acyclic (n=3) (*Figure 4A*). Although a stereotyped drop in the ratio of SEs$^{kiss}$ in the dark period normalized to those in the preceding light period was evident at the proestrus stage in young (reproductive) female mice (*Figure 1C*), no such trend was observed in all three groups of aging mice (*Figure 4B*), which suggests an alteration of negative feedback by sex steroid hormones. We also noticed a slight reduction in the frequency of SEs$^{kiss}$ in the irregular compared with the regular cyclic group (*Figure 4C*). In addition, in the irregular cyclic group, the number of SEs$^{kiss}$ in the light period was significantly lower than that in the dark period. These subtle differences were not evident in the chronic analysis of n=2 females (*Figure 3*), probably because of high data variations. Consistent with the individual analysis (*Figure 3D*), no difference in the cumulative probability of inter-peak intervals was found among

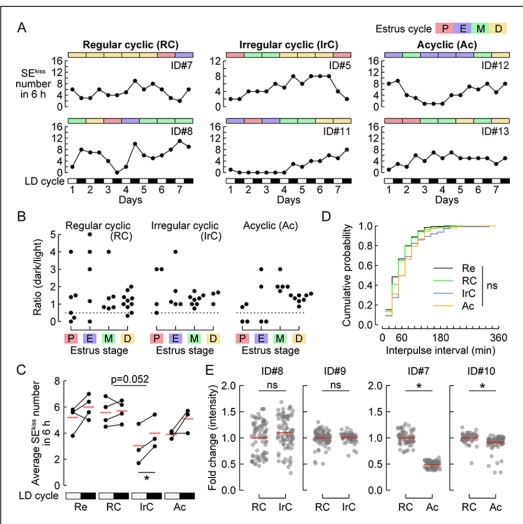

**Figure 4.** Characterization of synchronous episodes of ARC^kiss (SEs^kiss) in aging female mice. (**A**) Representative numbers of SEs^kiss in the estrus stages (color bars) for 7 days from regular cyclic, irregular cyclic, and acyclic female mice. SEs^kiss were recorded for 6 h in both light and dark periods. (**B**) The ratio of SEs^kiss in the dark period normalized to those in the preceding light period of regular cyclic (n=4), irregular cyclic (n=3), and acyclic (n=3) female mice, respectively. Dotted line = 0.5. (**C**) Average numbers of SEs^kiss in 6 h in the light (open boxes in the LD cycle) and dark (closed boxes in the LD cycle) periods of reproductive (Re), regular cyclic (RC), irregular cyclic (IrC), and acyclic (Ac) mice. Red horizontal lines indicate the mean. No interaction effect between estrus cyclicity and LD cycle was found by two-way ANOVA. *p<0.05 by paired *t*-test. Of note, the Re group (age 4–6 months) represents a reanalysis of the 5 days of data reported in *Figure 1* (corresponding to one estrus cycle). (**D**) Cumulative probability of inter-peak intervals among the Re (black), RC (green), IrC (blue), and Ac (orange) groups. ns: not significant by the Kolmogorov–Smirnov test. (**E**) Intensities of SEs^kiss during reproductive aging as assessed by the fold change of ΔF/F values in the IrC (#ID8, 9) and Ac (ID#7, 10) phases normalized to those in the RC phase. *p<0.05 by the Mann–Whitney *U* test. ns: not significant.

The online version of this article includes the following figure supplement(s) for figure 4:

**Figure supplement 1.** Histochemical analysis after imaging.

the reproductive group (age 4–6 months) or three aging groups (*Figure 4D*). Again, no increase in the frequency of SEs^kiss was observed in any of the three aging groups. Collectively, these data demonstrate that the frequency of SEs^kiss during the transition to reproductive senescence is overall stable.

We also analyzed whether the variability of the peak height would change in the aging female mice by assessing the coefficient of variance (*Figure 3—figure supplement 2C*). We noticed slight individual differences in the peak height variance in the regular and irregular cyclic aging groups; however, no significant inter-group differences were found. Regarding the intensities of SEs^kiss, our additional analysis (*Figure 4E*, and ID#5, 6 in *Figure 3E*) showed that the peak height declined from the regular to the irregular cyclic (two of four individuals) and acyclic phases (all four individuals). Therefore, our data suggest the notion that in aged female mice, the intensity of SEs^kiss may be reduced, whereas their frequency, to a lesser extent, appears altered.

Lastly, we assessed the expression level of *Kiss1*, *Tac2*, and *Pdyn* genes, which encode Kisspeptin, Neurokinin B, and Dynorphin, respectively, and collectively define ARC^kiss (KNDy) neurons, using quantitative PCR (qPCR) during reproductive senescence. The relative expression levels of these three genes normalized to that in the healthy phase, unaltered among the diestrus ARC samples from the reproductive to acyclic phase (*Figure 3—figure supplement 2D*). This result is in contrast to a previous report (*Kunimura et al., 2017*) implying a decrease in *Kiss1* expression in aged rats and supports the notion that a reduction in *Kiss1* gene expression is not a primary driver of the reproductive decline in mice. In addition, we conducted a post hoc histochemical analysis to investigate whether our long-term imaging of ARC^kiss damaged the target tissue. The targeting efficiency, as measured by the fraction of GCaMP6+ cells over *Tac2*+ cells, and the number of GCaMP6s+ cells were similar between healthy control mice in the reproductive phase and our aged samples collected after imaging sessions (*Figure 4—figure supplement 1B*). Moreover, the number of *Tac2*+ cells in the imaged side, normalized to the non-imaged contralateral side, was comparable between the control and recorded animals, suggesting the negligible tissue damage or cell death associated with cytotoxicity by prolonged expression of GCaMP6s and/ or repeated photometry recording (*Figure 4—figure supplement 1*).

# Discussion

Although reproductive aging significantly impacts the quality of life of women, its underlying neural mechanisms remain poorly characterized. We have established long-term chronic imaging of ARC[kiss] by fiber photometry in female mice from the fully reproductive phase to the end of the transition to reproductive senescent over 1 year. Our data illuminate the temporal dynamics of SEs[kiss] during the estrus cycle in the reproductive phase and during the transition to reproductive senescence. Our strategy of direct chronic visualization of hormonal regulators in the brain is generally applicable to facilitate characterizations of aging-associated robustness and/or malfunctions in each component of neuroendocrine systems. Here, we discuss specific insights and major limitations of the present study to guide the design of future investigations.

We updated the frequency and waveforms of individual SEs[kiss] in the young reproductive phase with a detailed analysis of the light/dark periods of all four stages of the estrus cycle (*Figure 1*). First, we reproduced a previous report showing a stereotyped drop of SEs[kiss] in response to a high level of progesterone just before ovulation in the proestrus stage (*Figure 1C*; *McQuillan et al., 2019*). Second, we found that the frequency of SEs[kiss] varied gradually depending on the stage of the estrus cycle (*Figure 1D*), in contrast to a bimodal frequency reported previously (*McQuillan et al., 2019*). Third, we noticed that the intensities of SEs[kiss], which have been thought to be constant across the estrus cycle (*McQuillan et al., 2019*), significantly varied during the estrus cycle (*Figure 1E*); individual peak heights of SEs[kiss] increased by almost 1.5-fold in the metestrus compared with the proestrus stage. This trend is consistent with the fact that the amplitude of the LH pulse is highest in the metestrus (*Czieselsky et al., 2016*). The enhanced activities of the GnRH pulse generator may be beneficial to provide LH/FSH more efficiently at the onset of follicular recruitment, which is sensitive to these hormones (*McGee and Hsueh, 2000*). What would be the underlying mechanism of these elevated activities? In one scenario, the number of ARC[kiss] that contributes to SEs[kiss] is constant, but individual ARC[kiss] fire more. Alternatively, more ARC[kiss] are dynamically recruited to the SEs[kiss] in the metestrus. To distinguish between these two possibilities, future studies using new tools that allow imaging or recording of neural activities at single-cell resolution (*Moore et al., 2022*; *Steinmetz et al., 2021*) in gonad-intact cycling female mice are needed.

We also found that the waveforms of individual SEs[kiss] varied depending on the stage of the estrus cycle. In the metestrus stage, when the peak was higher, the uphill was significantly steeper (*Figure 1F-H*). What would be the underlying mechanism? A recent Ca²⁺ imaging study of ARC[kiss] at single-cell resolution in ovariectomized female mice revealed heterogeneity of ARC[kiss] activations during each SE[kiss]: some subsets of ARC[kiss] act as 'leaders' and others as 'followers' (*Moore et al., 2022*). Given these temporal dynamics, in the metestrus stage, the number of 'leaders' may be increased to achieve quicker activation of the entire pool. Alternatively, the 'leaders' more efficiently activate 'followers' in the metestrus stage so that the latency to the peak is short. Although the exact mechanism remains an open question, our data suggest that the molecular and/or cellular mechanisms by which synchronized activation of the ARC[kiss] pool is achieved are dynamically modulated by the estrus stage, likely via feedback signals mediated by sex steroid hormones.

Our results also showed that the frequency of SEs[kiss] during reproductive senescence was generally stable. Classical studies suggest that the reduced reproductive functions in aging rodents are primarily the result of the dysfunction of the neuroendocrine system, independent of follicular depletion (*Kermath and Gore, 2012*; *Krohn, 1962*). There is a reduction in intensities of pulsatile LH release (*Scarbrough and Wise, 1990*; *Wise et al., 1988*) and kisspeptin expression in ARC[kiss] (*Kunimura et al., 2017*) in aged rats. A reduction in both the intensities and frequencies of pulsatile LH release is also known in postmenopausal women (*Hall et al., 2000*). In addition, a marked increase in the frequency of SEs[kiss] was observed in gonadectomized mice in the reproductive phase (*Clarkson et al., 2017*; *Han et al., 2019*) via a decline of sex steroid hormone-mediated feedback mechanisms. Based on these results, it was possible to assume an abnormal frequency of SEs[kiss] in older females; however, our data did not support this prediction. How is the relatively unaltered frequency of SEs[kiss] during the natural transition to reproductive senescence achieved? In one scenario, the levels of sex steroid hormones may be grossly intact, even in the acyclic phase (*Mobbs et al., 1985*; *Nilsson et al., 2015*). Alternatively, the reduction of sex steroid hormones in aging female mice (*Gee et al., 1983*; *Nelson et al., 1981*) may be counterbalanced by the attenuation of negative feedback mechanisms in ARC[kiss]. The lack of the stereotyped drop of SEs[kiss] during the proestrus stage in older females

(*Figure 4B*) generally supported a malfunction of negative feedback mechanisms by sex steroid hormones. It would be highly valuable to analyze SEs[kiss] patterns in aging ovariectomized animals to further investigate this matter, as the previous report of pulsatile LH secretion in aging ovariectomized rats (*Scarbrough and Wise, 1990*). Whatever the scenario, the lack of increased SEs[kiss] frequency in aged female mice might be adaptive in terms of thermoregulation because elevated SEs[kiss] are thought to be causal to vasodilation (flushing) (*Padilla et al., 2018*; *Rance et al., 2013*), one of the most commonly observed symptoms of perimenopausal disorders in women (*Casper et al., 1979*). In this sense, our data generally support the limitations in the use of older female mice as a model of the human menopausal transition, because not only the ovarian environment but also the adaptation of the hypothalamic regulators, may vary considerably between rodents and primates (*Gore, 2015*; *Kermath and Gore, 2012*).

Our data also suggest that the intensities of SEs[kiss], as assessed by the peak height, gradually declined from the regular to the acyclic phase (*Figure 3E*). Although we do not exclude the possibility that the reduction of photometric peaks was observed owing to technical artifacts, such as a slight misaligning of the optic fiber during long-term chronic imaging, we noticed that a drop in the photometric signals occurred in association with the reproductive phase rather than the total duration of fiber photometry. For instance, the peak intensity of ID#5 was relatively stable over 10 months from the reproductive to the irregular phase, but quickly declined in the subsequent 2 months upon the transition to the acyclic phase (*Figure 3A and E*). By contrast, ID#10 exhibited a quick drop of the peak intensity within just 3 months after the surgery upon entering into the acyclic phase. These observations support the interpretation that reduced photometric signals are associated with reduced neural activities of ARC[kiss]—individual firing rates, number of active neurons, or both. This trend can explain reduced LH amounts in each pulse (*Scarbrough and Wise, 1990*; *Wise et al., 1988*), leading to inefficient follicular development and steroidogenesis in the ovary. The activity decline of individual ARC[kiss] with the molecular underpinnings should be better characterized in future studies.

Besides our findings regarding SEs[kiss] dynamics, other factors may contribute to reproductive senescence. For instance, aging may alter the release probability of kisspeptin and its receptivity by GPR54 in the GnRH neurons. Additionally, aging affects the function of GnRH neurons (*Manfredi-Lozano et al., 2022*), including changes in their activity levels and the integrity of neural circuits with balanced excitatory and inhibitory inputs (*Kermath and Gore, 2012*). Furthermore, since ARC[kiss] is involved in restricting the amplitude of the preovulatory GnRH/LH surge (*Mittelman-Smith et al., 2016*), the absence of the stereotyped drop of SEs[kiss] during the proestrus stage in older females (*Figure 4B*) may contribute to the attenuation of the preovulatory GnRH/LH surge, leading to the transition to the acyclic phase. Future studies should examine these downstream processes to identify the cellular and molecular mechanisms by which the neuroendocrine functions are disrupted during the transition to reproductive senescence in female mice.

## Materials and methods
### Animals
All animal experiments were approved by the Institutional Animal Care and Use Committee of the RIKEN Kobe Branch. C57BL/6J female mice were purchased from Japan SLC (Shizuoka, Japan), and *Kiss1*-Cre::*GFP* (*Kiss1*-Cre) mice (Jax #017701) were purchased from the Jackson Laboratory. All animals were maintained at the animal facility of the RIKEN Center for Biosystems Dynamics Research (BDR) under an ambient temperature (18–23 °C) and a 12 h light/12 h dark cycle schedule. All mice had ad libitum access to a laboratory diet (MFG; Oriental Yeast, Shiga, Japan; 3.57 kcal/g) and water.

### Estrus cycle monitoring by vaginal cytology
To prevent the induction of pseudopregnancy, 10 μl of tap water was gently pipetted once or twice into the vagina with minimum insertion. These vaginal smears were placed on slides and evaluated in the types of cells under the microscope described below. The stage of the estrous cycle was determined based on the presence or absence of leukocytes and cornified and nucleated epithelial cells in accordance with a previous study (*Nelson et al., 1982*). Mostly nucleated and some cornified epithelial cells are present in proestrus. Some leukocytes are present if the female is in early proestrus. As the stage where the estrus cycle progresses from proestrus to estrus, mostly cornified epithelial cells

are present in a relatively larger amount of vaginal smears. These cornified epithelial cells with large amounts of leukocytes are observed in early metestrus. In late metestrus, there are some nucleated epithelial cells in large amounts of leukocytes. In diestrus, there is the lowest amount of vaginal smear that contains leukocytes dominantly and a few epithelial cells. An estrus cycle was defined as the period between subsequent proestrus stages, lasting for a minimum of four days and encompassing at least one day of estrus and at least one day of either metestrus or diestrus stages.

## Stereotaxic injection

For injection of AAV and an optical fiber, *Kiss1*-Cre mice were anesthetized with an intraperitoneal injection of saline containing 65 mg/kg ketamine (Daiichi Sankyo) and 13 mg/kg xylazine (X1251; Sigma-Aldrich) and head-fixed to a stereotaxic apparatus (RWD; #68045). The AAV driving Cre-dependent GCaMP6s (AAV9 *CAG-FLEx-GCaMPs-WPRE-SV40*, 1.5 × $10^{13}$ gp/ml; Addgene, #100842-AAV9) was injected into the ARC using the following coordinates from the bregma: anteroposterior (AP) = –2.0 mm, mediolateral = –0.2 mm, and dorsoventral (DV) = –5.9 mm. A total of 200 nL of AAV was injected into the ARC at a speed of 50 nl/min using a glass capillary regulated by a micro syringe pump (UMP3; World Precision Instruments). At 2–4 weeks after AAV injection, an optical fiber (numerical aperture = 0.50, core diameter = 400 μm; Kyocera) was placed over the ARC using the following coordinates from the bregma: AP = –1.8 mm, lateral = –0.2 mm, and DV = –5.7 mm defined on the brain atlas (*Franklin and Paxinos, 2013*). The injected fiber was fixed on the skull with dental cement (SUN MEDICAL, Shiga, Japan). Postsurgical mice were singly housed.

## Fiber photometry recording

Fluorescence signals were acquired using a fiber photometry system based on a previously published design (*Clarkson et al., 2017*; *Han et al., 2019*; *McQuillan et al., 2019*). All optical components were purchased from Doric Lenses (Quebec, Canada). We performed chronic $Ca^{2+}$ imaging by delivering excitation lights (470 nm modulated at 530.481 Hz) and collection of emitted fluorescence by using the integrated Fluorescence Mini Cube (iFMC4_IE(405)_E(460-490)_F(500-550)_S; Doric Lenses). Light collection, filtering, and demodulation were performed using the Doric photometry setup and Doric Neuroscience Studio Software (Doric Lenses). The power output at the tip of the fiber was about 4–40 μW. The signals were initially acquired at 12 kHz and then decimated to 120 Hz for recording to disk.

Calcium signals were analyzed using a custom-made code (see Source Code File) in R (version 4.0.3, http://www.R-project.org/). Because the signal intensity of SEs$^{kiss}$ varies widely owing to variations in surgery and individual variability, we first calculated the average ΔF/F height of the stereotyped SE$^{kiss}$ peak of each mouse from several visually obvious peaks with the full width at half maximum threshold over 10 s (*Clarkson et al., 2017*; *McQuillan et al., 2019*). The ΔF/F was calculated by $(F_t - F_0) / F_0$, where $F_t$ is the recorded signal at time = $t$ and $F_0$ is the average of signals over the entire 6 h of recording (*Figures 1A and 3A*). Then, SEs$^{kiss}$ were automatically detected by using the findpeaks function in R, with a peak height threshold of 40% of the estimated peak height. To analyze the reliable peak height and waveform further (*Figures 1E, 3E and 4E*), we extracted the raw data from –240 to +120 s, when the peak was set to 0 s. We then recalculated the ΔF/F by $(F_t - F_0) / F_0$, where $F_t$ is the recorded signal at time = $t$ and $F_0$ is the average of signals from –240 to –120 s. The FWQM, LWQM, and RWQM were determined by the recalculated ΔF/F value (*Figure 1H*). To show the averaged waveform (*Figure 1F and G*; *Figure 1—figure supplement 1B*, *Figure 3—figure supplement 2A, B*), the recalculated ΔF/F data were downsized at 10 Hz.

## Histochemistry

Mice were anesthetized with isoflurane and perfused with phosphate-buffered saline (PBS) followed by 4% paraformaldehyde (PFA) in PBS. The brain was post-fixed with 4% PFA in PBS overnight. Thirty-μm coronal brain sections (every fourth section) were made using a cryostat (Leica). To generate cRNA probes for *Tac2*, DNA templates were amplified by PCR from the whole-brain cDNA (Genostaff, cat#MD-01). T3 RNA polymerase recognition site (5'-*AATTAACCCTCACTAAAGGG*) was added to the 3' end of the reverse primers. Forward (F) and reverse (R) primers to generate DNA templates for cRNA probes are as follows: F 5'- *AGCCAGCTCCCTGATCCT*; R 5'-*TTGCTATGGGGGTTGAGGC*. DNA templates were subjected to in vitro transcription with DIG (cat#11277073910)-RNA labeling mix and T3 RNA polymerase (cat#11031163001) according to the manufacturer's instructions (Roche

Applied Science). Fluorescent in situ hybridization (ISH) combined with anti-GFP immunohistochemical staining was performed as previously reported (*Ishii et al., 2017*). In brief, after hybridization and washing, brain sections were incubated with horseradish peroxidase (HRP)-conjugated anti-Dig (Roche Applied Science cat#11207733910, 1:500) and anti-GFP (Aves Labs cat#GFP-1010, 1:500) antibodies overnight. Signals were amplified by TSA-plus Cyanine 3 (AKOYA Bioscience, NEL744001KT, 1:70 in 1×plus amplification diluent) for 25 min, followed by washing, and then GFP-positive cells were visualized by anti-chicken Alexa Fluor 488 (Jackson Immuno Research cat#703-545-155, 1:250). PBS containing 50 ng/ml 4',6-diamidino-2-phenylindole dihydrochloride (DAPI; Sigma-Aldrich, cat#D8417) was used for counter nuclear staining. Images were acquired using an Olympus BX53 microscope equipped with a 10x (N.A. 0.4) objective lens. Cells were counted manually.

In *Figure 4—figure supplement 1*, we prepared control animals by injecting AAV9 *CAG-FLEx-GCaMPs-WPRE-SV40* into the ARC of Kiss1-Cre female mice at 3–4 months of age. We sacrificed them 4 weeks following the surgery at 14 weeks of age, without fiber implantation or imaging. Thus, control samples represent the healthy reproductive phase without potential damage caused by repeated photometry recording and/or prolonged expression of GCaMP6s. The recorded samples were prepared by sacrificing aged animals (ranging from 18 to 29 months of age) after collecting photometry and estrus cycle data. We compared *Tac2* gene expression in the ARC as a robust marker of ARC$^{kiss}$ neurons because the expression level of *Kiss1* gene is not sufficient for stable detection by conventional in situ hybridization.

## Real-time quantitative PCR

Total RNA was extracted from the snap-frozen ARC tissues by using the TRIzol (Invitrogen) following the manufacturer's instructions. The cDNA from each sample was synthesized with oligo (deoxythymidine) primer at 37 °C by using the high-capacity cDNA reverse transcription kit (Applied Biosystems, cat#4368814). Gene expression levels were measured using the 7900 real-time PCR system (Applied Biosystems) with PowerUp SYBR Green Master Mix (Thermo Fisher Scientific Inc, cat #A25741). The primer sets used for *Kiss1*, *Tac2*, *Pdyn*, *Npy*, and *Gapdh* are as follows:

> *Kiss1*, 5'-TGCTGCTTCTCCTCTGT; 5'-ACCGCGATTCCTTTTCC
> *Tac2*, 5'-TGCTTCGGAGACTCTACGACAG; 5'-GTCCCACAAAGAAGTCGTGCATG
> *Pdyn*, 5'-CTGTGTGCAGTGAGGATTCAGG; 5'-GAGACCGTCAGGGTGAGAAAAG
> *Npy*, 5'-TACTCCGCTCTGCGACACTACA; 5'-GGCGTTTTCTGTGCTTTCCTTCA
> *Gapdh*, 5'- GCACCACCAACTGCTTAG; 5'- CAGTGATGGCATGGACTG

The specificity of the amplification products was confirmed by the dissociation curve analysis and electrophoresis on 1.5% agarose gels. The relative gene expression levels were calculated using the $2^{-\Delta\Delta Ct}$ method (*Livak and Schmittgen, 2001*), and were sequentially normalized first to *Gapdh* (a housekeeping gene) and then to *Npy* (an ARC regional marker). Finally, the expression data of each gene was normalized to the mean of the corresponding gene expression in the reproductive phase to generate the graph shown in *Figure 3—figure supplement 2D*.

## Statistical analysis

The statistical details of each experiment, including the statistical tests used, the exact value of *n*, and what *n* represents, are described in each figure legend. Two-way analysis of variance (ANOVA) was applied to confirm the interaction effects in *Figure 4C*. One-way repeated measures ANOVA followed by the Tukey–Kramer post hoc test was applied to determine significant differences in *Figure 1D, E and H*, and *Figure 1—figure supplement 1A*. One-way ANOVA followed by the Tukey–Kramer post hoc test was applied to determine significant differences in *Figures 2A–4C*. The Kruskal–Wallis test followed by the Mann–Whitney *U* test was applied to determine significant differences in *Figure 3E*, and *Figure 3—figure supplement 1B and C*, and 1D. The Mann–Whitney *U* test was applied to determine significant differences in *Figure 4E*, and *Figure 4—figure supplement 1B*. The Kolmogorov–Smirnov test was applied to determine significant differences in *Figures 3D and 4D*. Two-tailed Student's *t*-test was applied in *Figure 1G*. A paired *t*-test was applied in *Figure 4C*. All statistical analyses were performed using R. Differences were considered to be significant when the p-values were <0.05.

## Acknowledgements

We wish to thank RIKEN BDR animal facility staff for animal care, Satsuki Irie for technical support, Dr. Uenoyama (Nagoya University), Dr. Murata (the University of Tokyo), and the members of the Miyamichi Laboratory for the critical reading of the manuscript. This study was supported by Grants-in-Aid to TG (Nos. 19K16270 and 21K15194) and KM (Nos. 20K20589 and 21H02587) from the Japan Society for the Promotion of Science (JSPS). TG was also supported by the RIKEN Special Postdoctoral Researchers Program.

## Additional information

### Funding

| Funder | Grant reference number | Author |
|---|---|---|
| Japan Society for the Promotion of Science | 19K16270 | Teppei Goto |
| Japan Society for the Promotion of Science | 21K15194 | Teppei Goto |
| Japan Society for the Promotion of Science | 20K20589 | Kazunari Miyamichi |
| Japan Society for the Promotion of Science | 21H02587 | Kazunari Miyamichi |
| RIKEN | Special Postdoctoral Researchers Program | Teppei Goto |

The funders had no role in study design, data collection and interpretation, or the decision to submit the work for publication.

### Author contributions

Teppei Goto, Conceptualization, Data curation, Investigation, Methodology, Writing – original draft; Mitsue Hagihara, Investigation, Methodology; Kazunari Miyamichi, Conceptualization, Supervision, Methodology, Writing – original draft

### Author ORCIDs

Teppei Goto ⓘ http://orcid.org/0000-0002-0081-3357
Kazunari Miyamichi ⓘ http://orcid.org/0000-0002-7807-8436

### Ethics

All animal experiments were performed in strict accordance with the approved protocols by the Institutional Animal Care and Use Committee of the RIKEN Kobe Branch (Approval # A2017-15-12).

### Decision letter and Author response

Decision letter https://doi.org/10.7554/eLife.82533.sa1
Author response https://doi.org/10.7554/eLife.82533.sa2

## Additional files

### Supplementary files
- MDAR checklist
- Source code 1. A custom-made code in R for analyzing calcium signals.

### Data availability

Original fiber photometry data have been deposited in the SSBD: repository (https://ssbd.riken.jp/repository/297/). https://doi.org/10.24631/ssbd.repos.2023.04.297. The original code utilized in the course of this study is provided in Source Code File.

The following dataset was generated:

| Author(s) | Year | Dataset title | Dataset URL | Database and Identifier |
|---|---|---|---|---|
| Goto T, Miyamichi K | 2023 | Fiber photometry traces of the kisspeptin neurons in the arcuate nucleus of the hypothalamus in young and aging mice | https://ssbd.riken.jp/repository/297/ | doi.org/10.24631/ssbd.repos.2023.04.297, 10.24631/ssbd.repos.2023.04.297 |

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
