## [Editor Report]

The reviewers agreed that your studies demonstrating an age-dependent dynamic change in the activity of key neurons (Kisspeptin neurons) regulating reproductive functions will be of wide interest. The work provides functional links regarding the transition to reproductive senescence characteristic of aging. The use of chronic imaging of identified neurons is also an important advance.

---

## [Decision Letter]

**Decision letter after peer review:**

Thank you for submitting your article "Dynamics of Pulsatile Activities of Arcuate Kisspeptin Neurons in Aging Female Mice" for consideration by *eLife*. Your article has been reviewed by 2 peer reviewers, and the evaluation has been overseen by a Reviewing Editor and Catherine Dulac as the Senior Editor. The following individual involved in review of your submission has agreed to reveal their identity: Vincent Prévot (Reviewer #1).

Essential revisions:

1) It would strengthen the manuscript if the authors could discern from their chronic recordings whether the changes in kisspeptin population activity occur before the mouse estrous cycle transitions to irregular and acyclic phases.

2) Have the authors looked at kisspeptin expression in mice before and at the time when the change in kisspeptin activity was recorded?

3) The authors should define what parameters they have used to measure the beginning and end of a normal reproductive cycle. Is this when proestrus occurs? This may not align with the previous reporting of when reproductive senescence is reached in mice, in which the estrus cycles feature persistent vaginal cornification (PVC). In Figure 3, it appears mice do not enter PVC by the end of the recording period. This may suggest that the estrus cycle has been prolonged, but mice may not have reached acyclicity and reproductive senescence.

4) The mouse line is reported as Kiss1-Cre throughout the manuscript, but the authors have cited the Kiss1-CreGFP line. This should be corrected in the methods.

5) The authors have not conducted a histological analysis of kisspeptin cells after imaging. This would aid in (1) identifying the subset of kisspeptin cells imaged, (2) visualizing whether bleaching may have occurred, and (3) using immunocytochemical markers to confirm that cytotoxicity induced by repeated imaging has not affected the health and normal activity of the recorded cells.

6) Because reproductive aging does not alter markedly calcium dynamics in arcuate kisspeptin neurons, I would tone down the last sentence of the results (lines 212 and 213) and replace "generally support" by "suggest" and "is" by "may be" and "appears".

7) Regarding menopause, the authors should also mention in their introduction and/or discussion the paper of Hall and colleagues (PMID: 10843154) demonstrating the decrease in GnRH/LH pulse frequency with aging in postmenopausal women. Because the continuous infusion of kisspeptin has been shown to be able to restaure pulsatile LH release in women with hypothalamic amenorrhea (PMID: 24517142), pulsatile activity of arcuate nucleus kisspeptin neurons may not be required for GnRH/LH pulsatility, but its suppression in proestrus may play an important facilitatory role for the onset of the preovulatory GnRH/LH surge. The decrease of GnRH/LH pulsatile release with age could rather be due to a decrease or an alteration of the activity of the GnRH neurons themselves, as suggested by recent studies investigating GnRH neuronal function across aging in certain neurodevelopmental disorders (PMID: 36048943). These alternative interpretation and hypotheses, which do not hamper the importance of author's findings, must be discussed.

*Reviewer #1 (Recommendations for the authors):*

Because reproductive aging does not alter markedly calcium dynamics in arcuate kisspeptin neurons, I would tone down the last sentence of the results (lines 212 and 213) and replace "generally support" by "suggest" and "is" by "may be" and "appears".

To put these interesting results in the context of menopause, it might have been interesting to see the effect of ovariectomy in regular cyclic, irregular cyclic and acyclic animals on calcium dynamics (i.e., changes between before and after ovariectomy).

Regarding menopause, the authors should also mention in their introduction and/or discussion the paper of Hall and colleagues (PMID: 10843154) demonstrating the decrease in GnRH/LH pulse frequency with aging in postmenopausal women. Because the continuous infusion of kisspeptin has been shown to be able to restore pulsatile LH release in women with hypothalamic amenorrhea (PMID: 24517142), pulsatile activity of arcuate nucleus kisspeptin neurons may not be required for GnRH/LH pulsatility, but its suppression in proestrus may play an important facilitatory role for the onset of the preovulatory GnRH/LH surge. The decrease of GnRH/LH pulsatile release with age could rather be due to a decrease or an alteration of the activity of the GnRH neurons themselves, as suggested by recent studies investigating GnRH neuronal function across aging in certain neurodevelopmental disorders (PMID: 36048943). These alternative interpretations and hypotheses, which do not hamper the importance of author's findings, must be discussed.

*Reviewer #2 (Recommendations for the authors):*

It would strengthen the manuscript if the authors could discern from their chronic recordings whether the changes in kisspeptin population activity occur before the mouse estrous cycle transitions to irregular and acyclic phases.

Have the authors looked at kisspeptin expression in mice before and at the time when the change in kisspeptin activity was recorded? A reduction in kiss1 expression will support the conclusion in their discussion based on published literature.

The authors should define what parameters they have used to measure the beginning and end of a normal reproductive cycle. Is this when proestrus occurs? This may not align with the previous reporting of when reproductive senescence is reached in mice, in which the estrus cycles feature persistent vaginal cornification (PVC). In Figure 3, it appears mice do not enter PVC by the end of the recording period. This may suggest that the estrus cycle has been prolonged, but mice may not have reached acyclicity and reproductive senescence.

The mouse line is reported as Kiss1-Cre throughout the manuscript, but the authors have cited the Kiss1-CreGFP line. This should be corrected in the methods.

The authors have not conducted a histological analysis of kisspeptin cells after imaging. This would aid in (1) identifying the subset of kisspeptin cells imaged, (2) visualizing whether bleaching may have occurred, and (3) using immunocytochemical markers to confirm that cytotoxicity induced by repeated imaging has not affected the health and normal activity of the recorded cells.

---

## [Author Response]

Essential revisions:1) It would strengthen the manuscript if the authors could discern from their chronic recordings whether the changes in kisspeptin population activity occur before the mouse estrous cycle transitions to irregular and acyclic phases.

We greatly appreciate this insightful suggestion. We conducted an intensive re-analysis of all the data to find a parameter that correlates with the transitions from regular cyclic (RC) to irregular cyclic (IrC) and acyclic (Ac) phases in aging female mice. However, due to the limited sample size and large individual variations in our study, it turned out to be challenging to identify specific characteristics of ARC^kiss^ activities that can predict future transitions of estrus cyclicity. Our analysis revealed only one promising parameter, the right width at quarter maximum (RWQM) of SEs^kiss^ waveforms. We analyzed SEs^kiss^ waveforms in individual estrus stages and found that aging females in the RC phase displayed a reduction of RWQM values during the diestrus stage, suggesting a faster decay of ARC^kiss^ activities during SEs^kiss^ (Response Figure 1A). However, the RWQM values either increased or decreased further when the estrus cycle became IrC or Ac phase (Response Figure 1B). This finding suggests that individual variations of RWQM values during the diestrus phase increase upon reproductive senescence, a trait observed in 5 out of 6 animals we analyzed. As the predictability of this trait for estrus cycle transition requires future investigation, we regard this information as a preliminary observation that may be suitable only for this review forum.

**Author response image 1. sa2fig1:** Analysis of RWQM values during the diestrus stage in aging female mice. (A) Quantifications of RWQM of SE^kiss^ waveforms in two individual aging female mice (those analyzed in Figure 3). A typical decline in RWQM values is observed from reproductive (Re) to regular cyclic (RC) during aging, and then they increase again as the estrus cycle becomes irregular cyclic (IrC) or acyclic (Ac). Different letters (a–c, x–z) denote significant differences at P < 0.05 by the Kruskal–Wallis test followed by the Mann–Whitney U test. (B) The fold-change of RWQM values from RC to either IrC or Ac phase in four aging female mice (those analyzed in Figure 4). Unlike ID#5 and #6, these animals display a further reduction of RWQM values during the reproductive senescence. Although the origin of these individual variations remains unclear, our data show that 5 out of 6 aging animals exhibit significant changes in RWQM values during the diestrus phase associated with the transition from RC to IrC or Ac.

2) Have the authors looked at kisspeptin expression in mice before and at the time when the change in kisspeptin activity was recorded?

To address this important question, we assessed the expression level of *Kiss1*, *Tac2*, and *Pdyn* genes, which encode Kisspeptin, Neurokinin B, and Dynorphin, respectively, and collectively define ARC^kiss^ (KNDy) neurons, using quantitative PCR (qPCR) during reproductive senescence. We found that the relative expression levels of these three genes, normalized to that in the healthy phase, unaltered among the diestrus ARC samples from the reproductive to acyclic phase, which was reported in the new Figure 3—figure supplement 2D. This result supports the notion that a reduction in *Kiss1* gene expression may not be a primary driver of the reproductive decline in mice. We added an explanation of this result (Lines 200‒207) and related Methods (Lines 417‒435) in the revised manuscript.

3) The authors should define what parameters they have used to measure the beginning and end of a normal reproductive cycle. Is this when proestrus occurs? This may not align with the previous reporting of when reproductive senescence is reached in mice, in which the estrus cycles feature persistent vaginal cornification (PVC). In Figure 3, it appears mice do not enter PVC by the end of the recording period. This may suggest that the estrus cycle has been prolonged, but mice may not have reached acyclicity and reproductive senescence.

We also appreciate this important question. We clarified our definition of an estrus cycle in the Methods section (Lines 339‒341). We define an estrus cycle based on vaginal cytology that begins with proestrus and has a duration of 4 days or longer. Each cycle must include at least one day of the estrus stage and one day of the metestrus/diestrus stage, respectively.

Regarding PVC, we considered our mice ID#5 and #6 in Figure 3 to have entered acyclicity since they displayed persistent metestrus/diestrus stages in their 25-day vaginal smears, as illustrated in the new Figure 3—figure supplement 1. The persistent metestrus/diestrus stage is characteristic of reproductive senescence, which typically follows PVC. However, a previous study has reported that 25% of the mice can directly transition to the persistent metestrus/diestrus stage without PVC (Felicio LS et al. *Biology of Reproduction* 31, 446, 1984). Therefore, we speculate that we may have missed detecting the PVC due to our sampling interval, which spanned several months, or these mice may have skipped the PVC altogether.

4) The mouse line is reported as Kiss1-Cre throughout the manuscript, but the authors have cited the Kiss1-CreGFP line. This should be corrected in the methods.

According to this comment, we corrected the mouse information in the Methods section (Line 321).

5) The authors have not conducted a histological analysis of kisspeptin cells after imaging. This would aid in (1) identifying the subset of kisspeptin cells imaged, (2) visualizing whether bleaching may have occurred, and (3) using immunocytochemical markers to confirm that cytotoxicity induced by repeated imaging has not affected the health and normal activity of the recorded cells.

We concur with this comment and have conducted a post-hoc histochemical analysis. To robustly identify ARC^kiss^ neurons, we analyzed the expression of *Tac2* gene as a marker of ARC^kiss^ neurons, since the expression level of *Kiss1* gene is not sufficient for stable detection by conventional in situ hybridization. The targeting efficiency, as measured by the fraction of GCaMP6+ cells over *Tac2*+ cells, was comparable between healthy control mice in the reproductive phase and our aged samples collected after imaging sessions, as shown in the new Figure 4—figure supplement 1B. As we utilized an AAV to target GCaMP6s, the targeting efficiency is as expected. The number of GCaMP6s+ cells was comparable between the control and imaged samples, suggesting that there was no cellular loss in our aged and imaged samples. Moreover, the number of *Tac2*+ cells in the imaged side, normalized to the non-imaged contralateral side, was similar between the control and recorded animals, suggesting the negligible tissue damage or cell death associated with cytotoxicity resulting from prolonged expression of GCaMP6s and/or repeated photometry recording. We have included these data in the Results section (Lines 208‒216) and Figure 4—figure supplement 1, with related Methods (Lines 385‒415).

6) Because reproductive aging does not alter markedly calcium dynamics in arcuate kisspeptin neurons, I would tone down the last sentence of the results (lines 212 and 213) and replace "generally support" by "suggest" and "is" by "may be" and "appears".

We agree with this opinion and revised the text (Lines 197‒199).

7) Regarding menopause, the authors should also mention in their introduction and/or discussion the paper of Hall and colleagues (PMID: 10843154) demonstrating the decrease in GnRH/LH pulse frequency with aging in postmenopausal women. Because the continuous infusion of kisspeptin has been shown to be able to restaure pulsatile LH release in women with hypothalamic amenorrhea (PMID: 24517142), pulsatile activity of arcuate nucleus kisspeptin neurons may not be required for GnRH/LH pulsatility, but its suppression in proestrus may play an important facilitatory role for the onset of the preovulatory GnRH/LH surge. The decrease of GnRH/LH pulsatile release with age could rather be due to a decrease or an alteration of the activity of the GnRH neurons themselves, as suggested by recent studies investigating GnRH neuronal function across aging in certain neurodevelopmental disorders (PMID: 36048943). These alternative interpretation and hypotheses, which do not hamper the importance of author's findings, must be discussed.

We deeply appreciate these important suggestions and revised our Discussion (Lines 265‒267 and 304‒315).

Reviewer #1 (Recommendations for the authors):Because reproductive aging does not alter markedly calcium dynamics in arcuate kisspeptin neurons, I would tone down the last sentence of the results (lines 212 and 213) and replace "generally support" by "suggest" and "is" by "may be" and "appears".

See our Response #6.

To put these interesting results in the context of menopause, it might have been interesting to see the effect of ovariectomy in regular cyclic, irregular cyclic and acyclic animals on calcium dynamics (i.e., changes between before and after ovariectomy).

We concur with this view. While performing these experiments will undoubtedly provide valuable information, the preparation process will require a considerable amount of time. Consequently, we have highlighted these experiments as an important area for future studies in the Discussion (Line 278‒280).

Regarding menopause, the authors should also mention in their introduction and/or discussion the paper of Hall and colleagues (PMID: 10843154) demonstrating the decrease in GnRH/LH pulse frequency with aging in postmenopausal women. Because the continuous infusion of kisspeptin has been shown to be able to restore pulsatile LH release in women with hypothalamic amenorrhea (PMID: 24517142), pulsatile activity of arcuate nucleus kisspeptin neurons may not be required for GnRH/LH pulsatility, but its suppression in proestrus may play an important facilitatory role for the onset of the preovulatory GnRH/LH surge. The decrease of GnRH/LH pulsatile release with age could rather be due to a decrease or an alteration of the activity of the GnRH neurons themselves, as suggested by recent studies investigating GnRH neuronal function across aging in certain neurodevelopmental disorders (PMID: 36048943). These alternative interpretations and hypotheses, which do not hamper the importance of author's findings, must be discussed.

See our Response #7.

Reviewer #2 (Recommendations for the authors):It would strengthen the manuscript if the authors could discern from their chronic recordings whether the changes in kisspeptin population activity occur before the mouse estrous cycle transitions to irregular and acyclic phases.

See our response #1.

Have the authors looked at kisspeptin expression in mice before and at the time when the change in kisspeptin activity was recorded? A reduction in kiss1 expression will support the conclusion in their discussion based on published literature.

See our response #2.

The authors should define what parameters they have used to measure the beginning and end of a normal reproductive cycle. Is this when proestrus occurs? This may not align with the previous reporting of when reproductive senescence is reached in mice, in which the estrus cycles feature persistent vaginal cornification (PVC). In Figure 3, it appears mice do not enter PVC by the end of the recording period. This may suggest that the estrus cycle has been prolonged, but mice may not have reached acyclicity and reproductive senescence.

See our response #3.

The mouse line is reported as Kiss1-Cre throughout the manuscript, but the authors have cited the Kiss1-CreGFP line. This should be corrected in the methods.

See our Response #4.

The authors have not conducted a histological analysis of kisspeptin cells after imaging. This would aid in (1) identifying the subset of kisspeptin cells imaged, (2) visualizing whether bleaching may have occurred, and (3) using immunocytochemical markers to confirm that cytotoxicity induced by repeated imaging has not affected the health and normal activity of the recorded cells.

See our Response #5.